# The Ruminal Microbiome Alterations Associated with Diet-Induced Milk Fat Depression and Milk Fat Globule Size Reduction in Dairy Goats

**DOI:** 10.3390/ani14172614

**Published:** 2024-09-09

**Authors:** Menglu Zhang, Zhentao Liu, Kuixian Wu, Chuankai Zhang, Tong Fu, Yu Sun, Tengyun Gao, Liqiang Han

**Affiliations:** 1Henan International Joint Laboratory of Nutrition Regulation and Ecological Raising of Domestic Animal, College of Animal Science and Technology, Henan Agricultural University, Zhengzhou 450046, China; mlzhang0605@163.com (M.Z.); zhangck1998@163.com (C.Z.); futong2004@126.com (T.F.); sunyu@henau.edu.cn (Y.S.); 2Key Laboratory of Animal Biochemistry and Nutrition, Ministry of Agriculture and Rural Affairs, College of Verterinary Medicine, Henan Agricultural University, Zhengzhou 450046, China; 15890699466@139.com (Z.L.); 17739677771@163.com (K.W.)

**Keywords:** dairy goats, milk fat globule, milk fat globule size, ruminal bacteria

## Abstract

**Simple Summary:**

The aim of this study was to evaluate the effect of conjugated linoleic acid on the milk fat globule size and the ruminal microbiome of goats. In the present study, we found that dietary conjugated linoleic acid supplementation significantly and dose-dependently decreased the milk fat content and fat globule size in Saanen dairy goats, accompanied by an increase in the proportion of small-sized fat globules. Moreover, the dairy goats occurred in parallel with significant changes in the relative abundance of some ruminal bacterial populations. The milk fat content and fat globule size were correlated with the relative abundance of some bacteria, including members of Firmicutes and Bacteroidetes. Notably, our study was the first to analyze the correlation between the ruminal microbiota and fat globule size. These results constitute the first evidence for explaining the mechanism underlying diet-induced fat globules from the perspective of the ruminal microbiome in dairy goats.

**Abstract:**

The aim of this study was to evaluate the effect of conjugated linoleic acid (CLA) on milk fat globule (MFG) size and the ruminal microbiome of goats. Twenty-four mid-lactation Saanen dairy goats weighing 49 ± 4.5 kg (168 ± 27 d in milk, 1.2 ± 0.1 kg milk/d, 2–3 years old) were randomly divided into four groups—a control (CON) group, which was fed a basal diet, and three CLA supplementation groups, in which 30 g CLA (low-dose group, L-CLA), 60 g CLA (medium-dose group, M-CLA), or 90 g CLA (high-dose group, H-CLA) was added to the basal diet daily. The experiment lasted for 21 days, during which time goat milk was collected for composition and MFG size analysis. On day 21 of feeding, ruminal fluid was collected from the CON and H-CLA groups for analysis of the changes in microorganismal abundance. The results showed that CLA supplementation did not affect milk production, milk protein, or lactose content in the dairy goats (*p* > 0.05), but significantly reduced the milk fat content (*p* < 0.01) compared with the CON group. The CLA supplementation significantly decreased the D_[3,2]_ and D_[4,3]_ of the MFGs in a dose-dependent manner (*p* < 0.01). Moreover, dietary CLA inclusion increased the proportion of small-sized MFGs and decreased that of large-sized ones. The results of 16S rRNA gene sequencing showed that CLA-induced milk fat depression in dairy goats was accompanied by significant changes in the relative abundance of ruminal bacterial populations, most of which belonged to the Firmicutes and Bacteroidetes phyla. The relative abundance of *Rikenellaceae_RC9_gut_group* and *Prevolellaceae_UCG-003* in Bacteroidetes and *UCG-002*, *Succiniclasticum*, and norank_f__norank_o__*Clostridia_vadinBB60_*group in Firmicutes was significantly higher in the CON group than in the H-CLA group. In contrast, the relative abundance of norank_f__*UCG-011*, norank_f_*Eubacterium_coprostanoligenes*_group, unclassified_f__*Lachnospiraceae*, and *UCG-001* in Firmicutes and norank_f__*Muribaculaceae* in Bacteroidetes was significantly higher in the H-CLA group than in the CON group. Correlation analysis showed that the milk fat content was negatively correlated with the relative abundance of some bacteria, including members of Firmicutes and Bacteroidetes. Similarly, MFG size (D_[3,2]_ and D_[4,3]_) was negatively correlated with several members of Firmicutes and Bacteroidetes, including *Lachnospiraceae*, norank_f__*UCG-011*, *UCG-001*, norank_f__*Eubacterium_coprostanoligenes*_group (Firmicutes), and norank_f__*Muribaculaceae* (Bacteroidetes), while positively correlated with the relative abundance of some members of Firmicutes and Bacteroidetes, including *Mycoplasma*, *Succiniclasticum*, norank_f__norank_o__*Clostridia_vadinBB60_*group, *UCG-002* (Firmicutes), and *Rikenellaceae_RC9_gut_*group (Bacteroidetes). Overall, our data indicated that CLA treatment affected milk fat content and MFG size in dairy goats, and these effects were correlated with the relative abundance of ruminal bacterial populations. These results provide the first evidence to explain the mechanism underlying diet-induced MFG from the perspective of the ruminal microbiome in dairy goats.

## 1. Introduction

Goat milk is rich in nutrients and has unique medical and health-benefiting properties. Fat is the most variable main component in milk and its level can be affected by the ratio of dietary components [1]. In dairy cattle, it has been found that a high concentrate/low roughage feed or the addition of oil to the diet leads to a reduction in milk fat content and the occurrence of milk fat depression (MFD). Milk fat depression is a common phenomenon in ruminants that has puzzled breeders and scientists for over a century [2].

Fat in milk exists mainly in the form of milk fat globules (MFGs) [3]. The size of fat globules is an important characteristic of milk fat, affecting milk fat content and milk production and processing [4,5]. It has been reported that animal species and varieties, feeding methods, lactation stage, and seasons all affect the size of MFGs [6,7,8]. Conjugated linoleic acid (CLA) is an intermediate product of ruminal microbial metabolism in ruminants. The dietary addition of CLA can reduce the milk fat content of cows and goats, resulting in MFD [9,10,11]. Additionally, we previously found that CLA can reduce both the milk fat content of dairy cows, resulting in MFD, and the size of MFGs [12,13]. However, whether CLA affects the size of MFGs in dairy goats remains unclear.

Milk fat depression is a classic example of the interaction between dietary nutrients, gastrointestinal microbiota, and tissue physiology [14]. Dietary composition affects the structure of the ruminal microbiota [15]. Studies have found that ruminal bacteria are closely related to milk composition [16,17]. In large-scale dairy farming, the feeding of high-concentrate diets leads to a decline in milk fat content due to consequent changes in volatile fatty acid concentrations and ruminal microbiota composition [18]. Ruminal microbiota, as an important component of the digestive system of ruminants, is crucial for their early development, health, and physiology. Nevertheless, little is known about the role of ruminal microbiota in the regulation of MFG size.

Therefore, in this study, we hypothesized that microorganisms play a regulatory role in the MFD process in dairy goats. To prove this possibility, we first induced MFD in dairy goats through dietary CLA addition and subsequently assessed the effects on milk fat contents and MFG size. Additionally, we investigated the relationship between the ruminal microbiome and MFG in dairy goats, aiming to reveal the role of the gut microbiota in the MFD process in these ruminants.

## 2. Materials and Methods

### 2.1. Ethics Approval

The study was conducted on a dairy goat farm in central China. All procedures involving animals were performed in accordance with the experimental practices and standards approved by the Animal Welfare and Research Ethics Committee of Henan Agricultural University (HNND2024031807, 18 March 2024).

### 2.2. Animals, Diets, and Sampling

Twenty-four mid-lactation Saanen dairy goats weighing 49 ± 4.5 kg (1.2 ± 0.1 kg milk/d, 168 ± 27 d in milk, 2–3 years old)) were used in this study and were provided fresh feed twice a day at 06:00 and 16:00 h. Goats were housed in a naturally ventilated barn and fed individually; the diet was formulated to meet the nutritional requirements of dairy goats. The dietary information is listed in Appendix A. During the experiment, the goats were allowed free access to a total mixed ration and water. Through power analysis, it was calculated that 6 dairy goats per group were justified. The goats were randomly divided into four groups (*n* = 6/group)—a control (CON) group, which was fed a basal diet, and three CLA supplementation groups (a low-dose [L-CLA] group, a medium-dose [M-CLA] group and a high-dose [H-CLA] group), in which the animals were fed a basal diet containing 30 g CLA, 60 g CLA, and 90 g CLA daily, respectively. The whole experiment included a 5-d adaptation period and a 21-d feeding period. During the adaptation period, the amount of CLA added is gradually increased. Microencapsulated CLA powder was purchased from Qingdao Aohai Biological Co., Ltd., Qingdao, China. The fatty acid composition of the CLA is shown in Table 1. Milk production was recorded once daily. Raw milk samples were collected at 8:00 h every morning for the analysis of MFG size and milk composition. Some of the milk samples were sent to Henan DHI Testing Centre for the analysis of milk composition (including milk fat, protein, and lactose contents) using infrared spectrophotometry (Foss 120 Milko-Scan, Norborg, Denmark); some were sent back to the laboratory of Henan Agricultural University for MFG size measurement using a Mastersizer 3000 laser particle size analyzer (Malvern Instruments Ltd., Malvern, UK). Ruminal fluid was sampled from CON and H-CLA groups at the end of the 21-d feeding period and quickly stored in liquid nitrogen for microbiome analysis. The dairy goat was fixed with a neck clamp, the sampler pressed its tongue and tilted its head upward. Then, the disinfected ruminal fluid sampling tube was inserted into the mouth of the experimental animal, and the tube was sent to the throat along the tongue and through the esophagus. When the other end of the collection tube can obviously smell the odor of ruminal fluid and the experimental animal’s breathing is smooth, it indicates that the tube has reached the abdominal sac of the rumen. Next, the 100 mL syringe was connected to the end of the sampling tube, and the ruminal content could be extracted by pumping the syringe.

### 2.3. MFG Size Analysis

According to the operating instructions of the Mastersizer 3000, the refractive index was set to 1.560 for milk and 1.330 for pure water. A total of 400 mL of water was added to a beaker and stirred at 1500× *g*, after which approximately 1 mL of pure milk was added to the beaker. The system automatically obtained three measurements and calculated average particle size parameters. The values obtained for each sample were analyzed with Mastersizer software for the determination of the volume-related equivalent diameter (D_[4,3]_), surface area-related equivalent diameter (D_[3,2]_), and specific surface area (SSA) values.

### 2.4. Ruminal Microbial DNA Extraction and PCR Amplification

Total DNA was extracted from microbial communities using the E.Z.N.A. soil DNA kit (Omega Bio-tek, Norcross, GA, USA) according to the manufacturer’s instructions. The quality of the extracted DNA was assessed on a 1% agarose gel while DNA concentration and purity were determined with a NanoDrop 2000 UV–vis spectrophotometer (Thermo Scientific, Wilmington, DE, USA). The hypervariable V3–V4 region of the bacterial 16S rRNA gene was amplified on an ABI GeneAmp 9700 PCR thermocycler (ABI, Oakland, CA, USA) using the primer pair 338F (5′-ACTCCTACGGGAGGCAGCAG-3′) and 806R (5′-GGACTACHVGGGTWTCTAAT-3′). The PCR amplification conditions were as follows: an initial denaturation at 95 °C for 3 min, followed by 27 cycles of denaturing at 95 °C for 30 s, annealing at 55 °C for 30 s, and extension at 72 °C for 45 s, with a final extension at 72 °C for 10 min. The reaction mixture contained 4 μL of 5× TransStart FastPfu buffer, 2 μL of 2.5 mM dNTPs, 0.8 μL of forward primer (5 μM), 0.8 μL of reverse primer (5 μM), 0.4 μL of TransStart FastPfu DNA Polymerase, 10 ng of template DNA, and ddH_2_O to 20 μL. The PCR runs were performed in triplicate.

### 2.5. Illumina MiSeq Sequencing

The PCR products were extracted from 2% agarose gels and purified using the AxyPrep DNA Gel Extraction Kit (Axygen Biosciences, Union City, CA, USA) according to the manufacturer’s instructions and quantified using a Quantus Fluorometer (Promega, Madison, WI, USA). Purified amplicons were pooled in equimolar amounts and paired-end sequenced by Majorbio Bio-Pharm Technology Co. Ltd. (Shanghai, China) on an Illumina MiSeq PE300 platform/NovaSeq PE250 platform (Illumina, San Diego, CA, USA) according to standard protocols.

### 2.6. Processing of Sequencing Data

The raw 16S rRNA gene sequencing reads were demultiplexed, quality-filtered using fastp (v.0.20.0) [19], and merged using FLASH v.1.2.7 [19] based on the following criteria. (i) The 300-bp reads were truncated at any site receiving an average quality score of <20 over a 50-bp sliding window; truncated reads shorter than 50 bp and reads containing N bases were discarded. (ii) Only overlapping sequences longer than 10 bp were assembled according to their overlapping sequence. The maximum mismatch ratio for the overlapping region was 0.2; reads that could not be assembled were discarded. (iii) Samples were distinguished according to the barcoded primers, the sequence direction was adjusted, and the barcodes were compared for exact sequence matching; two nucleotide mismatches were allowed for primer pairing.

Operational taxonomic units (OTUs) were clustered at a 97% similarity threshold using UPARSE v.7.1 [19] and chimeric sequences were identified and removed. The taxonomy of each OTU representative sequence was analyzed by RDP Classifier version 2.2 [19] against the 16S rRNA database (eg. Silva v.138) using a confidence threshold of 70%.

### 2.7. Statistical Analysis

Milk yield, composition, and MFG size were analyzed using full factorial repeated measures of the general linear model (GLM) in SPSS 24 (SPSS, IBM Corp., Chicago, IL, USA). The alpha diversity was analyzed using a *t*-Test. The results are reported as means ± standard error of the mean. Correlations between microorganisms and MFG size were assessed using mstools software (https://mstools.shinyapps.io/shiny/, accessed on 20 June 2024). Statistical significance was set at *p* < 0.05.

## 3. Results

### 3.1. Effect of CLA on the Production Performance of Dairy Goats

As can be seen from Table 2, there were no differences in milk yield among the four groups (*p* > 0.05). Regarding milk composition, CLA treatment did not affect protein or lactose contents (*p* > 0.5) but significantly reduced the milk fat content in a dose-dependent manner (*p* < 0.01). Milk fat content decreased by 52% from 3.63 g/100 mL in the CON group to 1.74 g/100 mL in the H-CLA group.

### 3.2. The Effect of CLA on MFG Size Parameters

As shown in Figure 1, CLA supplementation reduced the size of MFGs in a dose-dependent manner (*p* < 0.01). Among the parameters, the D_[3,2]_ in the H-CLA and M-CLA groups (2.00 and 2.10 μm, respectively) was significantly lower than that in the CON group (3.33 μm) (*p* < 0.01). Meanwhile, the H-CLA group had the smallest D_[4,3]_ value (2.43 μm), followed by the M-CLA (2.78 μm) and the L-CLA (2.74 μm) groups, all of which were significantly lower than that of the CON group (3.41 μm). Additionally, CLA supplementation led to a greater MFG SSA (*p* < 0.01), which showed a trend of H-CLA group > M-CLA group > L-CLA group > CON group. That is to say, the addition of CLA significantly reduced the MFG size.

Further analysis of the proportions of fat globule sizes in goat milk revealed that compared with the CON group, the percentage of small-sized fat globules gradually increased, while that of large-sized fat globules gradually decreased with increasing CLA dosage (Figure 2, *p* < 0.05). For instance, the percentage of large-sized MFGs (3–5 μm) was approximately 5% in the H-CLA and M-CLA groups, 6% in the L-CLA group, and 8.8% in the CON group, while the percentage of small-sized MFGs (<1 μm) was approximately 11% in the H-CLA and M-CLA groups, 2.7% in the L-CLA group, and 1.5% in the CON group. Moreover, the percentage of very large-sized MFGs (>5 μm) also showed a decreased trend in the CLA group.

### 3.3. The Effect of CLA on the Composition of Ruminal Bacteria

16S rRNA gene sequencing analysis showed that a total of 3136 OTUs were shared by the CON and H-CLA groups of dairy goats (Figure 3). Rarefaction curve analysis demonstrated that the samples covered the majority of the ruminal bacteria (Appendix A). As can be seen from Figure 4, the bacterial richness index values (Sobs, ACE, and Chao) of the CLA supplementation group were significantly lower than those of the CON group (*p* < 0.05); however, no significant differences in the Shannon and Simpson indices were recorded between the two groups (Figure 4D,E; *p* > 0.1). The PCA plot showed a clear differentiation between the bacterial communities of the CON group and the H-CLA group, with PC1 explaining 25.59% and PC2 17.26% of the variation (Appendix A).

Taxonomic analysis identified a total of 20 phyla in the ruminal fluid of both groups, with 0.40% and 0.35% of the bacterial phyla remaining unclassified in the CON and H-CLA groups, respectively. At the genus level, meanwhile, a total of 364 genera were identified in the two groups (Figure 5). Bacteroidetes and Firmicutes were the dominant phyla in both groups of dairy goats, accounting for 94.0% and 94.6% of the relative abundance of the CON (Bacteroidetes 41.0%, Firmicutes 53.0%) and H-CLA (Bacteroidetes 48.9%, Firmicutes 45.7%) groups, respectively (Figure 5A). As shown in Figure 5B, *Prevotella* and *Rikenellaceae_RC9_gut_*group were the dominant genera in both groups. The relative abundance of *Prevotella* (Bacteroidetes) in the CON and H-CLA groups was 23.0% and 24.3%, respectively, and that of *Rikenellaceae_RC9_gut_*group (Bacteroidetes) was 11.25% and 6.2%, respectively. The average relative abundance of norank_f__*F082*, *Lachnospiraceae_NK3A20_*group, *Ruminococcus*, and *Suociniclasticum* was approximately 5% (5.85%, 5.2%, 4.85%, and 4.75%, respectively).

At the genus level, there were significant differences in the abundance of 21 bacterial genera between the H-CLA group and the CON group (Figure 6 shows the relative abundance of the top 15 genera), most of which belonged to the phyla Bacteroidetes and Firmicutes. As shown in Figure 6, the relative abundance of *Rikenellaceae_RC9_gut_*group (Bacteroidetes) and *UCG-002* (Firmicutes) in the CON group was almost twofold and threefold higher, respectively, than that in the H-CLA group. Meanwhile, the relative abundance of *Succiniclasticum* and norank_f__norank_o__*Clostridia_vadinBB60_*group in Firmicutes, *Prevolellaceae_UCG-003* in Bacteroidetes, and *Treponema* in Spirochaetota was significantly higher in the CON group than in the H-CLA group. Moreover, the relative abundance of norank_f__*UCG-011*, norank_f_*Eubacterium_coprostanoligenes*_group, unclassified_f__*Lachnospiraceae*, and *UCG-001* in Firmicutes and norank_f__*Muribaculaceae* in Bacteroidetes in the H-CLA group was significantly higher than that in the CON group.

### 3.4. Correlation of Differential Ruminal Bacteria with Goat Milk Composition and MFG Size

Spearman correlation analysis was used to explore the correlations of the goat milk composition and MFG size parameters with bacterial genera showing differential relative abundance between the CON and H-CLA groups of goats. As shown in Figure 7, milk fat content was negatively correlated with the relative abundance of unclassified_f__*Lachnospiraceae*, norank_f__*UCG-011*, and norank_f__*Eubacterium_coprostanoligenes*_group in Firmicutes; norank_f__*Muribaculaceae* in Bacteroidetes; and norank_f__norank_o__*Bradymonadales* in Desulfobacterota and positively correlated with the relative abundance of *Prevotellaceae_UCG-003* in Bacteroidetes, *UCG-002* in Firmicutes, and *Treponema* in Spirochaetota.

As depicted in Figure 8, MFG size (D_[3,2]_, D_[4,3]_,) was negatively correlated with the relative abundance of unclassified_f__*Lachnospiraceae*, norank_f__*UCG-011*, *UCG-001*, and norank_f__*Eubacterium_coprostanoligenes*_group in Firmicutes and norank_f__*Muribaculaceae* in Bacteroidetes and positively correlated with the relative abundance of *Mycoplasma*, *Succiniclasticum*, norank_f__norank_o__*Clostridia_vadinBB60_*group, and *UCG-002* in Firmicutes; *Treponema* in Spirochaetota; and *Rikenellaceae_RC9_gut_*group in Bacteroidetes. Meanwhile, the SSA was positively correlated with the relative abundance of norank_f__*UCG-011*, norank_f__*Muribaculaceae*, and norank_f__*Eubacterium_coprostanoligenes*_group, but negatively correlated with that of *Succiniclasticum*, norank_f__norank_o__*Clostridia_vadinBB60_*group, *UCG-002*, *Treponema*, *Rikenellaceae_RC9_gut_*group, *Pseudoramibacter,* and norank_f__*Bifidobacteriaceae*.

## 4. Discussion

Numerous studies have reported on the role of CLA in the occurrence of MFD. The CLA, an intermediate product of ruminal feed hydrogenation, engenders a decrease in milk fat content, a key factor leading to MFD in ruminants [20,21,22]. Baumgard et al. conducted a 5-day experiment involving the abomasal infusion of different doses of CLA in cows [23], Viswanadha et al. injected different doses of CLA directly into cows through the jugular vein for 5 days [24], and Zhang et al. added CLA to the feed of dairy cows and goats [25]; all found that CLA reduced the fat content in milk, but did not affect the milk protein, lactose content, or milk yield. Moreover, Zhang’s study has found that the milk fat content of both dairy cows and goats has decreased by about 50%. Our findings are in line with the results of these studies, that is, dietary CLA supplementation led to a dose-dependent reduction in the fat content of goat milk but did not affect milk yield, milk protein levels, or lactose content.

The size of MFG is closely related to lipids and proteins, which will affect its nutritional characteristics [26]. Smaller MFGs have higher milk fat globule membrane (MFGM) content [27]. MFGM indirectly affects the health of infants and adults. Compared to larger MFGs, smaller MFGs have higher concentrations of polar lipids. Sphingomyelin has been proven to have important biological functions in the development of the central nervous system in newborns [28]. Also, MFGM supplementation can improve the intestinal development of newborns [29]. It has been reported that MFG size can be affected by the nutritional composition of diets. Argov-Argaman et al. fed dairy cows a high-concentrate diet and found that the average size of MFG decreased from 3.51 to 3.3 μm [30]. Mesilati-Stahy et al. showed that the provision of a high-concentrate low-forage diet resulted in a smaller mean MFG diameter in cows with MFD [31]. We have previously found that dairy cows with CLA-induced MFD exhibit a significant change in milk fat content characterized by a decrease in the size of MFGs [12,13,32]. In the present study, we also found that dietary CLA inclusion led to a significant and CLA dose-dependent reduction in the size of MFGs in dairy goats, accompanied by an increase in the proportion of small-sized fat globules and a decrease in that of large-sized ones. Combined, the results of studies on cows and goats suggest that changes in fat globule size are the main characteristic of milk fat in ruminants with MFD.

Diet is known to affect the structure of ruminal microbial populations [33,34,35]. For instance, in the process of diet-induced MFD, alterations in ruminal fermentation led to changes in the biohydrogenation pathway, resulting in the inhibition of milk fat synthesis in the gland [36,37]. Diet-induced MFD is caused by highly fermentable diets rich in unsaturated fatty acids [38]. Elevated concentrations of polyunsaturated fatty acids, especially linoleic acid, can exert negative effects on the growth of ruminal bacteria [39,40], the main group of organisms known to carry out the biohydrogenation of unsaturated fatty acids in the rumen [40]. Although MFD is the result of microbial action, not all ruminal microbes involved in the biohydrogenation pathway are known. Pitta et al. detected a significant change in the ruminal bacterial spectrum in cows with diet-induced MFD and the bacterial population abundance and diversity were significantly reduced after 10 days of MFD induction [18]. In the present study, no significant differences in the Shannon and Simpson indices were found between the CON and CLA groups; however, the Sobs, ACE, and Chao index values were significantly lower in the H-CLA group than in the CON group. The higher the Shannon index or the smaller the Simpson index, the higher the diversity of the bacterial community; the higher the ACE and Chao indices, the richer the microbial community. Accordingly, our results indicated that CLA supplementation in the diet did not affect the diversity of ruminal bacteria in the goats but decreased the bacterial richness.

In the present study, Bacteroidetes and Firmicutes were the dominant phyla, which is consistent with the results reported for dairy cows [18,41]. Bacteroidetes, Firmicutes, and Fibrinobacteria decompose dietary carbohydrates, thereby indirectly providing energy for the body. Bacteroidetes are Gram-negative bacteria that mainly degrade non-fibrous substances such as starch, proteins, and polysaccharides [42]. In contrast, Firmicutes are Gram-positive bacteria and are the primary cellulose-degrading bacteria in the gastrointestinal tract [43]. *Prevotella* is the dominant genus among ruminal bacteria [44] and members of this group can decompose hemicellulose, proteins, and non-fibrous polysaccharides [45,46]. *Rikenellaceae_RC9_gut*_group is closely related to members of the Alipites family [47], and its function is to decompose plant-derived polysaccharides [48]. In the current study, we found that *Prevotella* and *Rikenellaceae_RC9_gut*_group (both belonging to Bacteroidetes) were the dominant genera. Liu et al. reported that *Prevotella 1* (Bacteroidetes) and *Succiniclasticum* (Firmicutes) were the dominant genera in two groups of cows with different total milk solid contents [41]. Huws et al. observed that the inclusion of flaxseed oil in the diet of steers increased the abundance of *Butyrivibrio*, *Howardella*, *Oribacterium*, *Pseudobutyrivibrio*, and *Roseburia*, whereas *Succinivibrio* and *Roseburia* were the only genera whose abundance was increased with echium oil supplementation [39]. This study explored the relative abundance of rumen bacterial populations with or without CLA supplements, which may enable us to link dietary nutrients, microbiota, and production parameters, thus understanding the role of ruminal bacteria in the MFD phenomenon.

The physiological status and milk composition of ruminants affect the composition of ruminal microorganisms. It was reported that bacterial abundance is lower in cows with high milk protein contents than in those with low milk protein yield [49,50]. Additionally, the abundance of *Bacteroides* in the rumen was reported to increase when cows entered the lactation stage and began to produce milk [51]. Zou et al. found a significant positive correlation between *Acinetobacter* and the average milk protein content [17], while Jami et al. showed that the relative abundance of *Prevotella* in the rumen of dairy cows was correlated with the fat content in milk [52]. Here, we found that the milk fat content was positively correlated with *Prevotellaceae_UCG-003*, *UCG-002*, and *Treponema,* indicating that the increase in CLA concentration in the diet may have a promoting effect on these bacteria. Whereas, it is negatively correlated with unclassified_f__*Lachnospiraceae*, norank_f__*UCG-011*, norank_f__*Eubacterium_coprostanoligenes*_group, norank_f__*Muribaculaceae*, and norank_f__norank_o__*Bradymonadales* in dairy goats.

Notably, our study was the first to analyze the correlation between the ruminal microbiota and MFG size. We observed that the D_[3,2]_ and D_[4,3]_ were negatively correlated with the relative abundance of norank_f__*UCG-011*, *UCG-001*, norank_f__*Eubacterium_coprostanoligenes*_group, and norank_f__*Muribaculaceae* and positively correlated with that of *Mycoplasma*, *Succiniclasticum*, norank_f__norank_o__*Clostridia_vadinBB60*_group, *UCG-002*, *Treponema*, *Rikenellaceae_RC9_gut*_group, and norank_f__*Bifidobacteriaceae*. This suggested that they play a role in altering the biohydrogenation pathway. The milk fat content and the size of the MFGs were positively correlated with the relative abundance of *Treponema*, *UCG-002*, and *Prevotellaceae_UCG-003* (Figure 7 and Figure 8) but exhibited a negative correlation with the relative abundance of norank_f__*Muribaculaceae*, norank_f__*UCG-011*, norank_f__*Eubacterium_coprostanoligenes*_group, and *UCG-001*. The CLA supplementation resulted in a correlation between milk fat content and MFG size [25], so there are bacteria that are consistent with the correlation between milk fat content and MFG size. In the current study, correlation analysis revealed the association between milk composition and MFG size with ruminal bacteria. However, it is unclear what the interactions among CLA, bacterial communities, and other dietary nutrients are. Future studies should investigate the effects of dietary CLA on specific bacterial populations in Bacteroidetes and Firmicutes for dietary recommendations.

## 5. Conclusions

In summary, in this work, we found that dietary CLA supplementation significantly and dose-dependently decreased the milk fat content and MFG size in Saanen dairy goats, accompanied by an increase in the proportion of small-sized MFGs and a decrease in that of large-sized MFGs. Moreover, dietary-induced MFD in dairy goats occurred in parallel with significant changes in the relative abundance of some ruminal bacterial populations. The milk fat content was negatively correlated with the relative abundance of some bacteria, including members of Firmicutes and Bacteroidetes. Meanwhile, MFG size was found to be positively correlated with more genera in the Bacteroidetes phylum and negatively correlated with more genera in the Firmicutes phylum. These results constitute the first evidence for explaining the mechanism underlying diet-induced MFD from the perspective of the ruminal microbiome in dairy goats.

## Figures and Tables

**Figure 1 animals-14-02614-f001:**
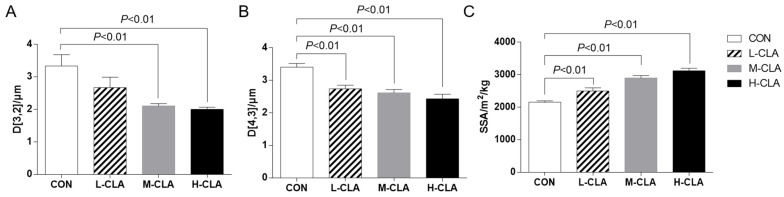
The effects of CLA on milk fat globule size. (**A**) D_[3,2]_: surface area-related equivalent diameter; (**B**) D_[4,3]_: volume-related equivalent diameter; (**C**) SSA: specific surface area. CON: control group; L-CLA: low-dose CLA group; M-CLA: medium-dose CLA group; H-CLA: high-dose CLA group.

**Figure 2 animals-14-02614-f002:**
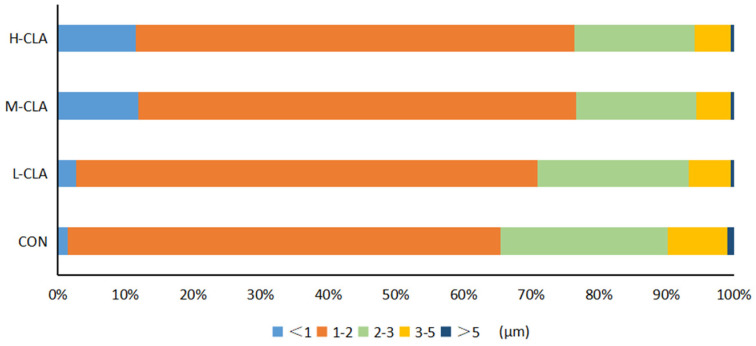
The effects of conjugated linoleic acid (CLA) on milk fat globule (MFG) size proportions. H-CLA: high-dose CLA group; M-CLA: medium-dose CLA group; L-CLA: low-dose CLA group; CON: control group.

**Figure 3 animals-14-02614-f003:**
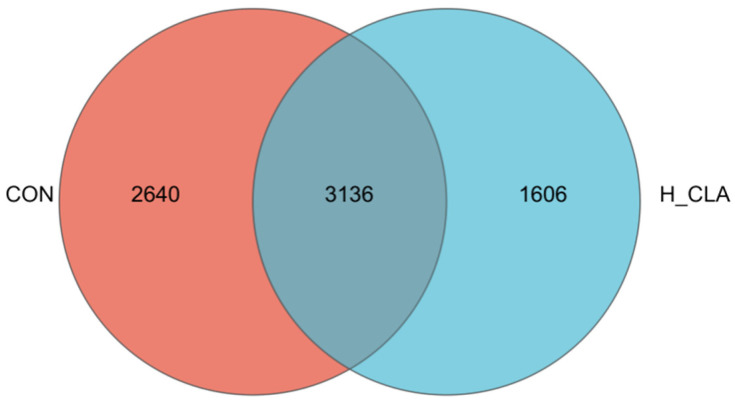
Shared and unique operational taxonomic units (OTUs) between the control (CON) and the high-dose conjugated linoleic acid (H-CLA) group.

**Figure 4 animals-14-02614-f004:**
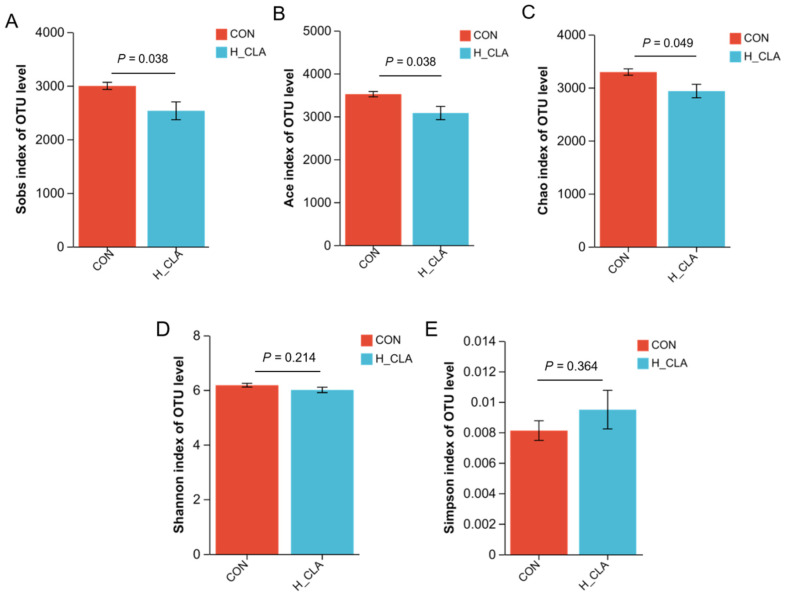
Comparative analysis of ruminal bacteria alpha-diversity between the control (CON) and high-dose conjugated linoleic acid (H-CLA) groups. (**A**) Sob index, (**B**) ACE index, (**C**) Chao index, (**D**) Shannon index, (**E**) Simpson index.

**Figure 5 animals-14-02614-f005:**
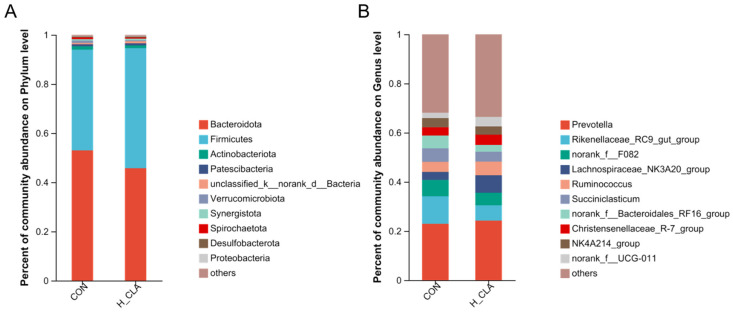
The relative abundance of ruminal bacteria at the phylum (**A**) and genus (**B**) levels in the control (CON) and high-dose conjugated linoleic acid group (H-CLA) groups.

**Figure 6 animals-14-02614-f006:**
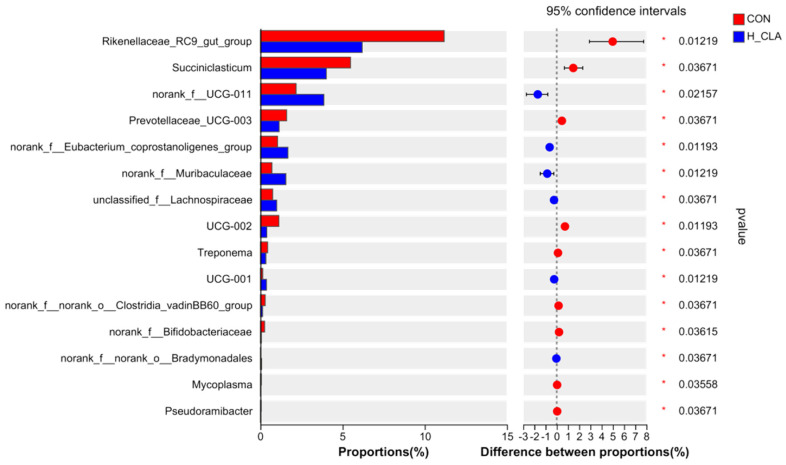
Comparison of the abundance of ruminal bacteria at the genus level between the control (CON) and high-dose conjugated linoleic acid (H-CLA) groups.

**Figure 7 animals-14-02614-f007:**
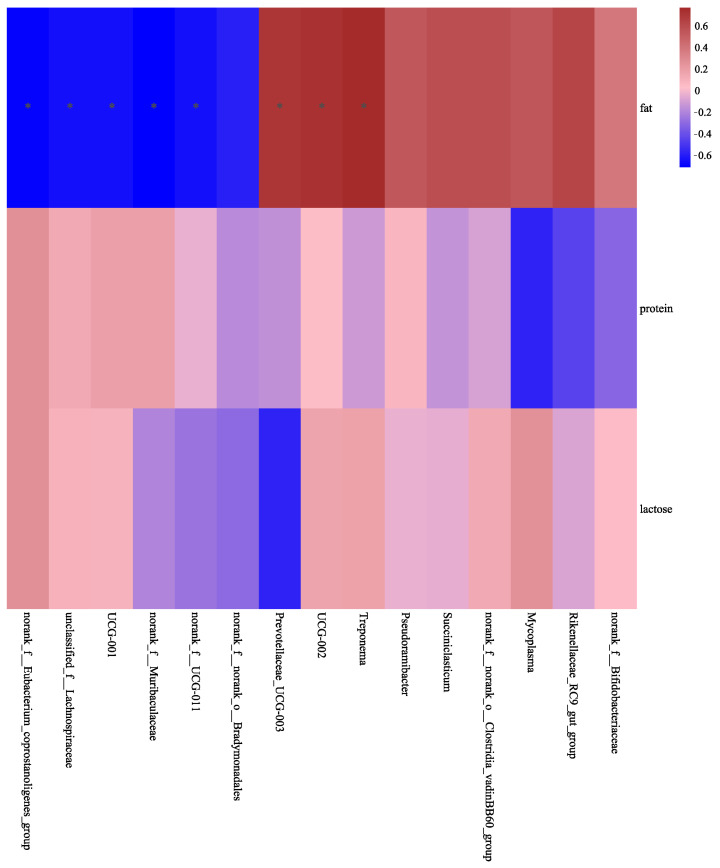
Correlations between differential bacteria and milk composition in the control (CON) and high-dose conjugated linoleic acid (H-CLA) groups. The color code indicates the direction of the correlations (red indicates a positive correlation and blue indicates a negative correlation); * *p* < 0.05.

**Figure 8 animals-14-02614-f008:**
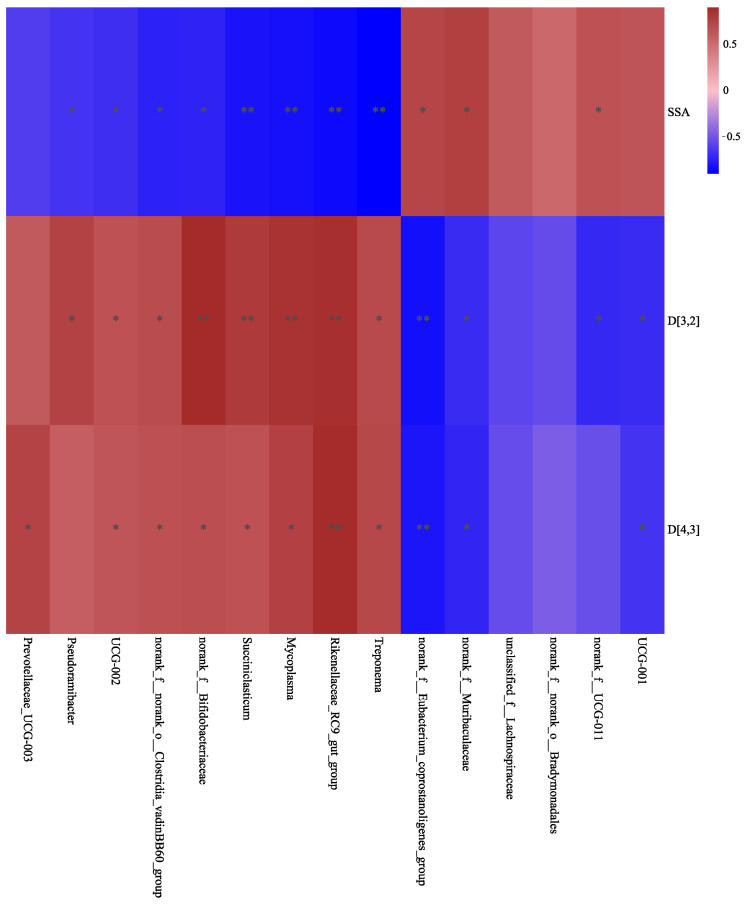
Correlations between differential bacteria and milk fat granule (MFG) size parameters in the control (CON) and high-dose conjugated linoleic acid (H-CLA) groups. The color code indicates the direction of the correlations (red indicates a positive and blue indicates a negative correlation); * *p* < 0.05 and ** *p* < 0.01.

**Table 1 animals-14-02614-t001:** Fatty acid composition of the conjugated linoleic acid.

Fatty Acid	Total Fatty Acids (%)
Palmitic acid	3.0
Stearic acid	2.7
Oleic acid	11.3
Linoleic acid	1.3
Conjugated linoleic acid	80.5
C18:2 *cis*-9, *trans*-11	38.5
C18:2 *trans*-10, *cis*-12	38.1
Other CLA isomers	3.8
Others	<0.1

CLA: conjugated linoleic acid.

**Table 2 animals-14-02614-t002:** The effects of CLA on milk production, milk composition, and MFG size.

Item	CON	L-CLA	M-CLA	H-CLA	SEM	*p*-Value
Milk yield (kg/day)	1.21	1.13	1.23	1.22	0.097	0.501
Milk fat (g/100 mL)	3.63 ^a^	2.69 ^b^	1.85 ^c^	1.74 ^c^	0.252	<0.01
Milk protein (g/100 mL)	3.52	3.54	3.48	3.79	0.193	0.676
Milk lactose (g/100 mL)	4.21	4.23	4.29	4.25	0.090	0.920

Note: In the same row, values with different small letter superscripts differ significantly (*p* < 0.05). CON: control group; L-CLA: low-dose conjugated linoleic acid group; M-CLA: medium-dose conjugated linoleic acid group; H-CLA: high-dose conjugated linoleic acid group; SEM: standard error of mean.

## Data Availability

The original contributions presented in the study are included in the article and Appendix A, further inquiries can be directed to the corresponding authors.

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
