# Peer review of "The Ruminal Microbiome Alterations Associated with Diet-Induced Milk Fat Depression and Milk Fat Globule Size Reduction in Dairy Goats"

_animals, 2024, doi:10.3390/ani14172614_

Round 1

Reviewer 1 Report

Comments and Suggestions for Authors

The manuscript is well-written and well-designed. The authors need to address a few comments.

line 2: in the title, please replace "MFG" with "milk fat globule"

line 15: delete "and a decrease in that of large-sized fat globule" 

lines 11-23: please try to shorten the simple summary, it should be 4-6 lines

line 25: what were the goats' ages and body weights?

line 25: please run a power test analysis to validate whether 6 animals per treatment is enough

line 25: Please mention the mid-lactation in days

line 29: the study period is a little bit short, do you think 21 days is enough to draw a strong recommendation

line 34: no need to mention any numbers in the abstract, please delete 

line 35, 63, 74, 290, 304, or elsewhere in the manuscript: don't start any sentence with an abbreviation

line 38: delete "(MFD)"

line 86: when the similarity report is checked, please try your best to minimize the similarity in this section 

line 91: Please provide the approval number and date

line 93: Please provide information on whether the kids stayed with the dams during the study, or they were already weaned

line 93: Please provide information about the housing system (whether the goats were penned as groups or in individual pens) 

line 179: delete "significant"

lines 180, 181, or elsewhere in the manuscript: delete "significantly"

line 182: delete numbers already mentioned in the tables or figures 

Author Response

Comments and Suggestions for Authors

The manuscript is well-written and well-designed. The authors need to address a few comments.

line 2: in the title, please replace "MFG" with "milk fat globule"

Response: We thank the reviewer for the comments. Corrections have been made in the revised version of line 2.

line 15: delete "and a decrease in that of large-sized fat globule" 

Response: We thank the reviewer for the comments. Corrections have been made in the revised version of line 15.

lines 11-23: please try to shorten the simple summary, it should be 4-6 lines

Response: We thank the reviewer for the comments. Corrections have been made in the revised version of line 11-21.

line 25: what were the goats' ages and body weights?

Response: We thank the reviewer for the comments. Corrections have been made in the revised version of line 24-25.

line 25: please run a power test analysis to validate whether 6 animals per treatment is enough

Response: We thank the reviewer for the comments. We have done many experiments and published papers (Xing et al., 2020, Zhang et al., 2021, Zhang et al., 2022,Zhang et al., 2024), these results confirm that whether cows or goats, 6 animals per treatment is enough to achieve the effect of CLA.

Reference

Xing Z Y,  Zhang M L, Wang Y Y, Yang G Y, Han L Q, & Loor J J. (2020). Short communication: A decrease in diameter of milk fat globules accompanies milk fat depression induced by conjugated linoleic acid supplementation in lactating dairy cows[J]. Journal of Dairy Science,  103(6): 5143-5147. https://doi.org/10.3168/jds.2019-17845

Zhang, M., T. Fu, Q. Huang, Z. Xing, J. Yang, W. Lu, M. Hu, L. Q. Han, J. J. Loor, and T. Y. Gao. 2022. Size, number and phospholipid composition of milk fat globules are affected by dietary conjugated linoleic acid. J Anim Physiol Anim Nutr 107: 995-1005.

Zhang, M. L., Z. Y. Xing, Q. X. Huang, and L. Q. Han. 2021. Effect of conjugated linoleic acid supplementation on fat globule size in raw milk. International Dairy Journal 115.

Zhang, Menglu; Liu, Zhentao; Kang, Fangyuan; Wu, Kuixian; Ni, Han; Han, Yingqian; Yang, Yanbin; Fu, Tong; Yang, Guoyu; Gao, Tengyun; Han, Liqiang.(2024). Food chemistry, 439, 138101 DOI 10.1016/j.foodchem.2023.138101,

line 25: Please mention the mid-lactation in days

Response: We thank the reviewer for the comments. Corrections have been made in the revised version of line 24-25.

line 29: the study period is a little bit short, do you think 21 days is enough to draw a strong recommendation

Response: We thank the reviewer for the comments. As for the treatment time, Bauman (2008) has indicated that the effects of CLA isomer on milk fat reaches a nadir by 4 to 5 d of supplementation and dependent on CLA dose. In our previous studies, dairy cows feeding with CLA for 6 days (Xing et al., 2020), 8 days (Zhang et al., 2021), 14 days (Zhang et al., 2022) and dairy goats feeding with CLA for 21 days (Zhang et al., 2024) all had a significantly decline of milk fat and size of MFG. So we select 21 days as the feeding time to obtain successful MFD.

Reference

Bauman, D. E., J. W. Perfield, K. J. Harvatine, and L. H. Baumgard. 2008. Regulation of fat synthesis by conjugated linoleic acid: Lactation and the ruminant model. J Nutr 138(2):403-409.

line 34: no need to mention any numbers in the abstract, please delete 

Response: We thank the reviewer for the comments. Corrections have been made in the revised version of abstract.

line 35, 63, 74, 290, 304, or elsewhere in the manuscript: don't start any sentence with an abbreviation

Response: We thank the reviewer for the comments. Corrections have been made in the revised version of line 33, 62, 75, 301 and 391.

line 38: delete "(MFD)"

Response: We thank the reviewer for the comments. Corrections have been made in the revised version of line 36.

line 86: when the similarity report is checked, please try your best to minimize the similarity in this section 

Response: We thank the reviewer for the comments. Corrections have been made in the revised MS.

line 91: Please provide the approval number and date

Response: We thank the reviewer for the comments. Corrections have been made in the revised version of line 97.

line 93: Please provide information on whether the kids stayed with the dams during the study, or they were already weaned

Response: We thank the reviewer for the comments. The kids have been weaned and they were not in the same pen with the dams.

line 93: Please provide information about the housing system (whether the goats were penned as groups or in individual pens) 

Response: We thank the reviewer for the comments. Corrections have been made in the revised version of line 101-102.

line 179: delete "significant"

Response: We thank the reviewer for the comments. Corrections have been made in the revised version of line 190.

lines 180, 181, or elsewhere in the manuscript: delete "significantly"

Response: We thank the reviewer for the comments. Corrections have been made in the revised version of line 200.

line 182: delete numbers already mentioned in the tables or figures 

Response: We thank the reviewer for the comments. Corrections have been made in the revised version of line 193-194.

Reviewer 2 Report

Comments and Suggestions for Authors

Comments: The authors investigate the effect of CLA supplementation on the size of milk fat globule, and the relative abundance of ruminal microbiome in dairy goats. Although the topic of the manuscript is of interest and particular importance in the field, there are still inaccuracies. Some suggestions are made here.

Specific comments

L90: Please provide additional ethic information.

L99: Supplement Table A1: is the ingredient and nutritional composition of the diet based on dry matter or not? Please clarify. Is the nutritional content calculated or measured? There is no description in the article.

L109: How many times a day are the goats milked?

L116: Frozen milk sample on the 21st day is used for what?

L193: “protein and lactose” should be changed to “protein or lactose”.

     (P > 0.5) should be followed by “,”.

L212: “globule” is spelling mistake.

L222: The very large-sized MFGs (>5 μm) did not showed a significantly dose-dependent manner.

L226: Materials and Methods 2.2 did not indicate that there were only two groups of ruminal bacteria.

L370: What does “also” mean?

L389-401: This paragraph compares dairy cows to goats. There are differences between the two animals, is this comparison reasonable?

Comments on the Quality of English Language

Minor editing of English language required

Author Response

L90: Please provide additional ethic information.

Response: We thank the reviewer for the comments. Corrections have been made in the revised version of line 97.

L99: Supplement Table A1: is the ingredient and nutritional composition of the diet based on dry matter or not? Please clarify. Is the nutritional content calculated or measured? There is no description in the article.

Response: We thank the reviewer for the comments. Corrections have been made in  Supplement Table A1.

L109: How many times a day are the goats milked?

Response: We thank the reviewer for the comments. The goats were milked once a day in the morning.

L116: Frozen milk sample on the 21st day is used for what?

Response: We thank the reviewer for the comments. Corrections have been made in the revised version of line 119.

L193: “protein and lactose” should be changed to “protein or lactose”.

     (P > 0.5) should be followed by “,”.

Response: We thank the reviewer for the comments. Corrections have been made in the revised version of line 191-192.

L212: “globule” is spelling mistake.

Response: We thank the reviewer for the comments. Corrections have been made in the revised version of line 216.

L222: The very large-sized MFGs (>5 μm) did not showed a significantly dose-dependent manner.

Response: We thank the reviewer for the comments. Corrections have been made in the revised version of line 226-227.

L226: Materials and Methods 2.2 did not indicate that there were only two groups of ruminal bacteria.

Response: We thank the reviewer for the comments. Corrections have been made in the revised version of line 119-120.

L370: What does “also” mean?

Response: We thank the reviewer for the comments. Corrections have been made in the revised version of line 364.

L389-401: This paragraph compares dairy cows to goats. There are differences between the two animals, is this comparison reasonable?

Response: We thank the reviewer for the comments. Dairy Cows and dairy goats are ruminants, they both have rumen and mammary gland. Many studies of CLA have been conducted on cows and dairy goats, so it is reasonable to compare the two animals.

Reviewer 3 Report

Comments and Suggestions for Authors

The manuscript presents a well-conducted study on the effects of conjugated linoleic acid (CLA) supplementation on milk fat globule (MFG) size and the ruminal microbiome in Saanen dairy goats. The findings are significant, contributing to the understanding of diet-induced milk fat depression (MFD) and its association with alterations in the ruminal microbiome. The study is thorough in its design, including detailed analysis of milk composition, MFG size, and microbiome alterations, and it addresses a relevant issue in dairy science.

However, while the study is comprehensive, the manuscript could benefit from clearer presentation and further discussion in several areas. The results are robust, but the manuscript would be strengthened by more explicit connections between the findings and their implications for dairy goat management. Additionally, some methodological details and statistical analyses require clarification.

1. The title is descriptive but could be more concise. Consider revising it to something like "Effects of CLA Supplementation on Milk Fat Globule Size and Ruminal Microbiome in Dairy Goats."

2. The abstract is well-written but could include a more explicit statement of the study's objectives and key findings. Ensure that the significance of the findings is clearly conveyed.

3. The introduction provides a solid background but could benefit from a clearer statement of the research gap. Clearly articulate how this study builds on previous research and what specific questions it aims to answer.

4. While the design is sound, more details on the selection criteria for the dairy goats and how they were allocated to the different CLA dosage groups would improve the transparency of the study.

5. The statistical methods used should be described in more detail, particularly regarding the handling of potential confounding factors. Clarify whether any adjustments were made for multiple comparisons.

6. The results section is detailed, but it could benefit from a clearer presentation of the key findings. Consider using bullet points or a summary table to highlight the most important results related to milk composition.

7. The discussion of MFG size is comprehensive, but the manuscript would be improved by including visual aids, such as histograms or box plots, to illustrate the distribution of MFG sizes across the different groups.

8. The microbiome analysis is thorough, but it would be helpful to include a more detailed discussion of the biological significance of the observed changes in specific bacterial genera. Explain how these changes might contribute to MFD.

9. The discussion does a good job of linking the results to the broader context, but it could be more explicit in drawing connections between the microbiome changes and MFD. Consider discussing the potential mechanisms in more detail.

10. The practical implications of the findings should be more clearly articulated. Discuss how these results could inform dairy goat management practices, particularly in relation to diet formulation.

Author Response

  1. The title is descriptive but could be more concise. Consider revising it to something like "Effects of CLA Supplementation on Milk Fat Globule Size and Ruminal Microbiome in Dairy Goats."

Response: We thank the reviewer for the comments. At the beginning, our title was indeed “Effects of CLA Supplementation on Milk Fat Globule Size and Ruminal Microbiome in Dairy Goats”, but this title could not well reflect the relationship between rumen microorganisms and milk fat, especially MFG, so the title was finally changed to “The Ruminal Microbiome Alterations Associated with Diet-induced Milk Fat Depression and Milk Fat Globule Size Reduction in Dairy Goats”.

  1. The abstract is well-written but could include a more explicit statement of the study's objectives and key findings. Ensure that the significance of the findings is clearly conveyed.

Response: We thank the reviewer for the comments. Corrections have been made in the revised version of line 52-56.

  1. The introduction provides a solid background but could benefit from a clearer statement of the research gap. Clearly articulate how this study builds on previous research and what specific questions it aims to answer.

Response: We thank the reviewer for the comments. Corrections have been made in the revised version of line 75-85.

  1. While the design is sound, more details on the selection criteria for the dairy goats and how they were allocated to the different CLA dosage groups would improve the transparency of the study.

Response: We thank the reviewer for the comments. Corrections have been made in the revised version of line 99-103.

  1. The statistical methods used should be described in more detail, particularly regarding the handling of potential confounding factors. Clarify whether any adjustments were made for multiple comparisons.

Response: We thank the reviewer for the comments. We did not make any adjustments using multiple comparisons. Corrections have been made in Statistical analysis.

  1. The results section is detailed, but it could benefit from a clearer presentation of the key findings. Consider using bullet points or a summary table to highlight the most important results related to milk composition.

Response: We thank the reviewer for the comments. Corrections have been made in the revised version of line 193-194, 207-208.

  1. The discussion of MFG size is comprehensive, but the manuscript would be improved by including visual aids, such as histograms or box plots, to illustrate the distribution of MFG sizes across the different groups.

Response: We thank the reviewer for the comments. Corrections have been made in the Figure 1.

  1. The microbiome analysis is thorough, but it would be helpful to include a more detailed discussion of the biological significance of the observed changes in specific bacterial genera. Explain how these changes might contribute to MFD.

Response: We thank the reviewer for the comments. Corrections have been made in the revised version of discussion.

  1. The discussion does a good job of linking the results to the broader context, but it could be more explicit in drawing connections between the microbiome changes and MFD. Consider discussing the potential mechanisms in more detail.

Response: We thank the reviewer for the comments. Corrections have been made in the revised version of discussion.

  1. The practical implications of the findings should be more clearly articulated. Discuss how these results could inform dairy goat management practices, particularly in relation to diet formulation.

Response: We thank the reviewer for the comments. Corrections have been made in the revised version of discussion.
